# Online to Offline Conversions, Universality and Adaptive Minibatch Sizes

**Kfir Y. Levy**
Department of Computer Science, ETH Zürich.
`yehuda.levy@inf.ethz.ch`

## Abstract

We present an approach towards convex optimization that relies on a novel scheme which converts adaptive online algorithms into offline methods. In the offline optimization setting, our derived methods are shown to obtain favourable adaptive guarantees which depend on the *harmonic sum* of the queried gradients. We further show that our methods implicitly adapt to the objective's structure: in the smooth case fast convergence rates are ensured without any prior knowledge of the smoothness parameter, while still maintaining guarantees in the non-smooth setting. Our approach has a natural extension to the stochastic setting, resulting in a lazy version of SGD (stochastic GD), where minibathces are chosen *adaptively* depending on the magnitude of the gradients. Thus providing a principled approach towards choosing minibatch sizes.

## 1 Introduction

Over the past years *data adaptiveness* has proven to be crucial to the success of learning algorithms. The objective function underlying "big data" applications often demonstrates intricate structure: the scale and smoothness are often unknown and may change substantially in between different regions/directions, [1]. Learning methods that acclimatize to these changes may exhibit superior performance compared to non adaptive procedures.

State-of-the-art first order methods like AdaGrad, [1], and Adam, [2], adapt the learning rate on the fly according to the feedback (i.e. gradients) received during the optimization process. AdaGrad and Adam are guaranteed to work well in the *online* convex optimization setting, where loss functions may be chosen *adversarially* and change between rounds. Nevertheless, this setting is harder than the stochastic/offline settings, which may better depict practical applications. Interestingly, even in the offline convex optimization setting it could be shown that in several scenarios very simple schemes may substantially outperform the output of AdaGrad/Adam. An example of such a simple scheme is choosing the point with the smallest gradient norm among all rounds. In the first part of this work we address this issue and design adaptive methods for the offline convex optimization setting. At heart of our derivations is a novel scheme which converts adaptive online algorithms into offline methods with favourable guarantees[1]. Our shceme is inspired by standard online to batch conversions, [3].

A seemingly different issue is choosing the minibatch size, $b$, in the stochastic setting. Stochastic optimization algorithms that can access a noisy gradient oracle may choose to invoke the oracle $b$ times in every query point, subsequently employing an averaged gradient estimate. Theory for stochastic convex optimization suggests to use a minibatch of $b = 1$, and predicts a degradation of $\sqrt{b}$ factor upon using larger minibatch sizes[2]. Nevertheless in practice larger minibatch sizes are usually found to be effective. In the second part of this work we design stochastic optimization methods in

which minibatch sizes are chosen *adaptively* without any theoretical degradation. These are natural extensions of the offline methods presented in the first part.

Our contributions:

**Offline setting:** We present two (families of) algorithms AdaNGD (Alg. 2) and SC-AdaNGD (Alg. 3) for the convex/strongly-convex settings which achieve favourable adaptive guarantees (Thms. 2.1, 2.2, 3.1, 3.2 ). The latter theorems also establish their universality, i.e., their ability to implicitly take advantage of the objective's smoothness and attain rates as fast as GD would have achieved if the smoothness parameter was known. In contrast to other universal approaches such as line-search-GD, [4], and universal gradient [5], we do so *without* any line search procedure. Concretely, without the knowledge of the smoothness parameter our algorithm ensures an $O(1/\sqrt{T})$ rate in general convex case and an $O(1/T)$ rate if the objective is also smooth (Thms. 2.1, 2.2). In the strongly-convex case our algorithm ensures an $O(1/T)$ rate in general and an $O(\exp(-\gamma T))$ rate if the objective is also smooth (Thm. 3.2 ), where $\gamma$ is the condition number.

**Stochastic setting:** We present Lazy-SGD (Algorithm 4) which is an extension of our offline algorithms. Lazy-SGD employs larger minibatch sizes in points with smaller gradients, which selectively reduces the variance in the "more important" query points. Lazy-SGD guarantees are comparable with SGD in the convex/strongly-convex settings (Thms. 4.2, 4.3).

On the technical side, our online to offline conversion schemes employ three simultaneous mechanisms: an adaptive online algorithm used in conjunction with gradient normalization and with a respective importance weighting. To the best of our knowledge the combination of the above techniques is novel, and we believe it might also find use in other scenarios.

This paper is organized as follows. In Sections 2,3, we present our methods for the offline convex/strongly-convex settings. Section 4 describes our methods for the stochastic setting, and Section 5 concludes. Extensions and a preliminary experimental study appear in the Appendix.

## 1.1 Related Work

The authors of [1] simultaneously to [6], were the first to suggest AdaGrad—an adaptive gradient based method, and prove its efficiency in tackling online convex problems. AdaGrad was subsequently adjusted to the deep-learning setting to yield the RMSprop, [7], and Adadelta, [8], heuristics. Adam, [2], is a popular adaptive algorithm which is often the method of choice in deep-learning applications. It combines ideas from AdaGrad together with momentum machinery, [9].

An optimization procedure is called universal if it implicitly adapts to the objective's smoothness. In [5], universal gradient methods are devised for the general convex setting. Concretely, without the knowledge of the smoothness parameter, these methods attain the standard $O(1/T)$, an accelerated $O(1/T^2)$ rates for smooth objectives, and an $O(1/\sqrt{T})$ rate in the non-smooth case. The core technique in this work is a line search procedure which estimates the smoothness parameter in every iteration. For strongly-convex and smooth objectives, line search techniques, [4], ensure linear convergence rate, *without* the knowledge of the smoothness parameter. However, line search is not "fully universal", in the sense that it holds no guarantees in the non-smooth case. For the latter setting we present a method which is "fully universal" (Thm. 3.2), nevertheless it *requires* the strong-convexity parameter.

The usefulness of employing normalized gradients was demonstrated in several non-convex scenarios. In the context of quasi-convex optimization, [10], and [11], established convergence guarantees for the offline/stochastic settings. More recently, it was shown in [12], that normalized gradient descent is more appropriate than GD for saddle-evasion scenarios.

In the context of stochastic optimization, the effect of minibatch size was extensively investigated throughout the past years, [13, 14, 15, 16, 17, 18]. Yet, all of these studies: **(i)** assume a smooth expected loss, **(ii)** discuss fixed minibatch sizes. Conversely, our work discusses adaptive minibatch sizes, and applies to both smooth/non-smooth expected losses.

## 1.2 Preliminaries

**Notation:** $\|\cdot\|$ denotes the $\ell_2$ norm, $G$ denotes a bound on the norm of the objective's gradients, and $[T] := \{1, \ldots, T\}$. For a set $\mathcal{K} \in \mathbb{R}^d$ its diameter is defined as $D = \sup_{x,y \in \mathcal{K}} \|x - y\|$. Next we

---
**Algorithm 1** Adaptive Gradient Descent (AdaGrad)
---
  **Input**: #Iterations $T$, $x_1 \in \mathbb{R}^d$, set $\mathcal{K}$
  Set: $Q_0 = 0$
  **for** $t = 1 \dots T$ **do**
    Calculate: $g_t = \nabla f_t(x_t)$
    Update:
$$Q_t = Q_{t-1} + \|g_t\|^2$$

    Set: $\eta_t = D/\sqrt{2Q_t}$
    Update: $x_{t+1} = \Pi_{\mathcal{K}}\left(x_t - \eta_t g_t\right)$
  **end for**
---

define $H$-strongly-convex/$\beta$-smooth functions,

$$f(y) \geq f(x) + \nabla f(x)^\top (y - x) + \frac{H}{2}\|x - y\|^2; \quad \forall x, y \in \mathcal{K} \quad \text{(\textit{H}-strong-convexity)}$$

$$f(y) \leq f(x) + \nabla f(x)^\top (y - x) + \frac{\beta}{2}\|x - y\|^2; \quad \forall x, y \in \mathcal{K} \quad \text{(\textit{β}-smoothness)}$$

### 1.2.1 AdaGrad

The methods presented in this paper lean on AdaGrad (Alg. 1), an online optimization method which employs an adaptive learning rate. The following theorem states AdaGrad's guarantees, [1],

**Theorem 1.1.** *Let $\mathcal{K}$ be a convex set with diameter $D$. Let $\{f_t\}_{t=1}^T$ be an arbitrary sequence of convex loss functions. Then Algorithm 1 guarantees the following regret;*

$$\sum_{t=1}^T f_t(x_t) - \min_{x \in \mathcal{K}} \sum_{t=1}^T f_t(x) \leq \sqrt{2D^2 \sum_{t=1}^T \|g_t\|^2} \ .$$

## 2 Adaptive Normalized Gradient Descent (AdaNGD)

In this section we discuss the convex optimization setting and introduce our AdaNGD$_k$ algorithm, which depends on a parameter $k \in \mathbb{R}$. We first derive a general convergence rate which holds for a general $k$. Subsequently, we elaborate on the $k = 1, 2$ cases which exhibit universality as well as adaptive guarantees that may be substantially better compared to standard methods.

Our method AdaNGD$_k$ is depicted in Alg. 2. This algorithm can be thought of as an *online to offline conversion scheme* which utilizes AdaGrad (Alg. 1) as a black box and eventually outputs a weighted sum of the online queries. Indeed, for a fixed $k \in \mathbb{R}$, it is not hard to notice that AdaNGD$_k$ is equivalent to invoking AdaGrad with the following loss sequence $\{\tilde{f}_t(x) := g_t^\top x/\|g_t\|^k\}_{t=1}^T$. And eventually weighting each query point inversely proportional to the $k$'th power norm of its gradient. The reason behind this scheme is that in offline optimization it makes sense to dramatically reduce the learning rate upon uncountering a point with a very small gradient. For $k \geq 1$, this is achieved by invoking AdaGrad with gradients normalized by their $k$'th power norm. Since we discuss constrained optimization, we use the projection operator defined as, $\Pi_{\mathcal{K}}(y) := \min_{x \in \mathcal{K}} \|x - y\|$ . The next lemma states the guarantee of AdaNGD for a general $k$:

**Lemma 2.1.** *Let $k \in \mathbb{R}$, $\mathcal{K}$ be a convex set with diameter $D$, and $f$ be a convex function; Also let $\bar{x}_T$ be the output of AdaNGD$_k$ (Algorithm 2), then the following holds:*

$$f(\bar{x}_T) - \min_{x \in \mathcal{K}} f(x) \leq \frac{\sqrt{2D^2 \sum_{t=1}^T 1/\|g_t\|^{2(k-1)}}}{\sum_{t=1}^T 1/\|g_t\|^k}$$

*Proof sketch.* Notice that the AdaNGD$_k$ algorithm is equivalent to applying AdaGrad to the following loss sequence: $\{\tilde{f}_t(x) := g_t^\top x/\|g_t\|^k\}_{t=1}^T$. Thus, applying Theorem 1.1, and using the definition of $\bar{x}_T$ together with Jensen's inequality the lemma follows. $\square$

---

**Algorithm 2** Adaptive Normalized Gradient Descent (AdaNGD$_k$)

---

**Input**: #Iterations $T$, $x_1 \in \mathbb{R}^d$, set $\mathcal{K}$, parameter $k$
Set: $Q_0 = 0$
**for** $t = 1 \ldots T-1$ **do**
    Calculate: $g_t = \nabla f(x_t)$, $\hat{g}_t = g_t/\|g_t\|^k$
    Update:
$$Q_t = Q_{t-1} + 1/\|g_t\|^{2(k-1)}$$
    Set $\eta_t = D/\sqrt{2Q_t}$
    Update: $x_{t+1} = \Pi_\mathcal{K}(x_t - \eta_t \hat{g}_t)$
**end for**
**Return**: $\bar{x}_T = \sum_{t=1}^{T} \frac{1/\|g_t\|^k}{\sum_{\tau=1}^{T} 1/\|g_\tau\|^k} x_t$

---

For $k = 0$, Algorithm 2 becomes AdaGrad (Alg. 1). Next we focus on the cases where $k = 1, 2$, showing improved adaptive rates and universality compared to GD/AdaGrad. These improved rates are attained thanks to the adaptivity of the learning rate: when query points with small gradients are encountered, AdaNGD$_k$ (with $k \geq 1$) reduces the learning rate, thus focusing on the region around these points. The hindsight weighting further emphasizes points with smaller gradients.

## 2.1 AdaNGD$_1$

Here we show that AdaNGD$_1$ enjoys a rate of $O(1/\sqrt{T})$ in the non-smooth convex setting, and a fast rate of $O(1/T)$ in the smooth setting. We emphasize that the same algorithm enjoys these rates simultaneously, without any prior knowledge of the smoothness or of the gradient norms.

From Algorithm 2 it can be noted that for $k = 1$ the learning rate becomes independent of the gradients, i.e. $\eta_t = D/\sqrt{2t}$, the update is made according to the direction of the gradients, and the weighting is inversely proportional to the norm of the gradients. The following Theorem establishes the guarantees of AdaNGD$_1$,

**Theorem 2.1.** *Let $k = 1$, $\mathcal{K}$ be a convex set with diameter $D$, and $f$ be a convex function; Also let $\bar{x}_T$ be the outputs of AdaNGD$_1$ (Alg. 2), then the following holds:*

$$f(\bar{x}_T) - \min_{x \in \mathcal{K}} f(x) \leq \frac{\sqrt{2D^2 T}}{\sum_{t=1}^{T} 1/\|g_t\|} \leq \frac{\sqrt{2}GD}{\sqrt{T}} .$$

*Moreover, if $f$ is also $\beta$-smooth and the global minimum $x^* = \arg\min_{x \in \mathbb{R}^n} f(x)$ belongs to $\mathcal{K}$, then:*

$$f(\bar{x}_T) - \min_{x \in \mathcal{K}} f(x) \leq \frac{D\sqrt{T}}{\sum_{t=1}^{T} 1/\|g_t\|} \leq \frac{4\beta D^2}{T} .$$

*Proof sketch.* The data dependent bound is a direct corollary of Lemma 2.1. The general case bound holds by using $\|g_t\| \leq G$. The bound for the smooth case is proven by showing $\sum_{t=1}^{T} \|g_t\| \leq O(\sqrt{T})$. This translates to a lower bound $\sum_{t=1}^{T} 1/\|g_t\| \geq \Omega(T^{3/2})$, which concludes the proof. $\square$

The data dependent bound in Theorem 2.1 may be substantially better compared to the bound of the GD/AdaGrad. As an example, assume that half of the gradients encountered during the run of the algorithm are of $O(1)$ norms, and the other gradient norms decay proportionally to $O(1/t)$. In this case the guarantee of GD/AdaGrad is $O(1/\sqrt{T})$, whereas AdaNGD$_1$ guarantees a bound that behaves like $O(1/T^{3/2})$. Note that the above example presumes that all algorithms encounter the same gradient magnitudes, which might be untrue. Nevertheless in the smooth case AdaNGD$_1$ provably benefits due to its adaptivity.

## 2.2 AdaNGD$_2$

Here we show that AdaNGD$_2$ enjoys comparable guarantees to AdaNGD$_1$ in the general/smooth case. Similarly to AdaNGD$_1$ the same algorithm enjoys these rates simultaneously, without any prior knowledge of the smoothness or of the gradient norms. The following Theorem establishes the guarantees of AdaNGD$_2$,

**Algorithm 3** Strongly-Convex AdaNGD (SC-AdaNGD$_k$)

---

**Input**: #Iterations $T$, $x_1 \in \mathbb{R}^d$, set $\mathcal{K}$, strong-convexity $H$, parameter $k$
Set: $Q_0 = 0$
**for** $t = 1 \dots T - 1$ **do**
    Calculate: $g_t = \nabla f(x_t)$, $\hat{g}_t = g_t / \|g_t\|^k$
    Update:
$$Q_t = Q_{t-1} + 1/\|g_t\|^k$$
    Set $\eta_t = 1/HQ_t$
    Update: $x_{t+1} = \Pi_{\mathcal{K}} (x_t - \eta_t \hat{g}_t)$
**end for**
**Return**: $\bar{x}_T = \sum_{t=1}^{T} \frac{1/\|g_t\|^k}{\sum_{\tau=1}^{T} 1/\|g_\tau\|^k} x_t$

---

**Theorem 2.2.** *Let $k = 2$, $\mathcal{K}$ be a convex set with diameter $D$, and $f$ be a convex function; Also let $\bar{x}_T$ be the outputs of AdaNGD$_2$ (Alg. 2), then the following holds:*

$$f(\bar{x}_T) - \min_{x \in \mathcal{K}} f(x) \le \frac{\sqrt{2D^2}}{\sqrt{\sum_{t=1}^{T} 1/\|g_t\|^2}} \le \frac{\sqrt{2}GD}{\sqrt{T}} \; .$$

*Moreover, if $f$ is also $\beta$-smooth and the global minimum $x^* = \arg\min_{x \in \mathbb{R}^n} f(x)$ belongs to $\mathcal{K}$, then:*

$$f(\bar{x}_T) - \min_{x \in \mathcal{K}} f(x) \le \frac{\sqrt{2D^2}}{\sqrt{\sum_{t=1}^{T} 1/\|g_t\|^2}} \le \frac{4\beta D^2}{T} \; .$$

It is interesting to note that AdaNGD$_2$ will have always performed better than AdaGrad, had both algorithms encountered the same gradient norms. This is due to the well known inequality between arithmetic and harmonic means, [19], $\frac{1}{T} \sum_{t=1}^{T} a_t \ge \frac{1}{\frac{1}{T} \sum_{t=1}^{T} 1/a_t}$, $\forall \{a_t\}_{t=1}^{T} \subset \mathbb{R}_+$, which directly implies, $\frac{1}{\sqrt{\sum_{t=1}^{T} 1/\|g_t\|^2}} \le \frac{1}{T} \sqrt{\sum_{t=1}^{T} \|g_t\|^2}$ .

## 3 Adaptive NGD for Strongly Convex Functions

Here we discuss the offline optimization setting of strongly convex objectives. We introduce our SC-AdaNGD$_k$ algorithm, and present convergence rates for general $k \in \mathbb{R}$. Subsequently, we elaborate on the $k = 1, 2$ cases which exhibit universality as well as adaptive guarantees that may be substantially better compared to standard methods.

Our SC-AdaNGD$_k$ algorithm is depicted in Algorithm 3. Similarly to its non strongly-convex counterpart, SC-AdaNGD$_k$ can be thought of as an online to offline conversion scheme which utilizes an online algorithm which we denote SC-AdaGrad (we elaborate on the latter in the appendix). The next Lemma states its guarantees,

**Lemma 3.1.** *Let $k \in \mathbb{R}$, and $\mathcal{K}$ be a convex set. Let $f$ be an $H$-strongly-convex function; Also let $\bar{x}_T$ be the outputs of SC-AdaNGD$_k$ (Alg. 3), then the following holds:*

$$f(\bar{x}_T) - \min_{x \in \mathcal{K}} f(x) \le \frac{1}{2H \sum_{t=1}^{T} \|g_t\|^{-k}} \sum_{t=1}^{T} \frac{\|g_t\|^{-2(k-1)}}{\sum_{\tau=1}^{t} \|g_\tau\|^{-k}} \; .$$

*Proof sketch.* In the appendix we present and analyze SC-AdaGrad. This is an *online* first order algorithm for strongly-convex functions in which the learning rate decays according to $\eta_t = 1/\sum_{\tau=1}^{t} H_\tau$, where $H_\tau$ is the strong-convexity parameter of the loss function at time $\tau$. Then we show that SC-AdaNGD$_k$ is equivalent to applying SC-AdaGrad to the following loss sequence:

$$\left\{ \tilde{f}_t(x) = \frac{1}{\|g_t\|^k} g_t^\top x + \frac{H}{2\|g_t\|^k} \|x - x_t\|^2 \right\}_{t=1}^{T} \; .$$

The lemma follows by combining the regret bound of SC-AdaGrad together with the definition of $\bar{x}_T$ and with Jensen's inequality. $\square$

For $k = 0$, SC-AdaNGD becomes the standard GD algorithm which uses learning rate of $\eta_t = 1/Ht$. Next we focus on the cases where $k = 1, 2$.

## 3.1 SC-AdaNGD$_1$

Here we show that SC-AdaNGD$_1$ enjoys a rate of $\tilde{O}(1/T)$ for strongly-convex objectives, and a faster rate of $\tilde{O}(1/T^2)$ assuming that the objective is also smooth. We emphasize that the same algorithm enjoys these rates simultaneously, without any prior knowledge of the smoothness or of the gradient norms. The following theorem establishes the guarantees of SC-AdaNGD$_1$,

**Theorem 3.1.** *Let $k = 1$, and $\mathcal{K}$ be a convex set. Let $f$ be a $G$-Lipschitz and $H$-strongly-convex function; Also let $\bar{x}_T$ be the outputs of SC-AdaNGD$_1$ (Alg. 3), then the following holds:*

$$ f(\bar{x}_T) - \min_{x \in \mathcal{K}} f(x) \leq \frac{G\left(1 + \log\left(\sum_{t=1}^{T} \frac{G}{\|g_t\|}\right)\right)}{2H\sum_{t=1}^{T}\frac{1}{\|g_t\|}} \leq \frac{G^2(1 + \log T)}{2HT} . $$

*Moreover, if $f$ is also $\beta$-smooth and the global minimum $x^* = \arg\min_{x \in \mathbb{R}^n} f(x)$ belongs to $\mathcal{K}$, then,*

$$ f(\bar{x}_T) - \min_{x \in \mathcal{K}} f(x) \leq \frac{(\beta/H)G^2\left(1 + \log T\right)^2}{HT^2} . $$

## 3.2 SC-AdaNGD$_2$

Here we show that SC-AdaNGD$_2$ enjoys the standard $\tilde{O}(1/T)$ rate for strongly-convex objectives, and a linear rate assuming that the objective is also smooth. We emphasize that the same algorithm enjoys these rates simultaneously, without any prior knowledge of the smoothness or of the gradient norms. In the case where $k = 2$ the guarantee of SC-AdaNGD is as follows,

**Theorem 3.2.** *Let $k = 2$, $\mathcal{K}$ be a convex set, and $f$ be a $G$-Lipschitz and $H$-strongly-convex function; Also let $\bar{x}_T$ be the outputs of SC-AdaNGD$_2$ (Alg. 3), then the following holds:*

$$ f(\bar{x}_T) - \min_{x \in \mathcal{K}} f(x) \leq \frac{1 + \log(G^2\sum_{t=1}^{T}\|g_t\|^{-2})}{2H\sum_{t=1}^{T}\|g_t\|^{-2}} \leq \frac{G^2(1 + \log T)}{2HT} . $$

*Moreover, if $f$ is also $\beta$-smooth and the global minimum $x^* = \arg\min_{x \in \mathbb{R}^n} f(x)$ belongs to $\mathcal{K}$, then,*

$$ f(\bar{x}_T) - \min_{x \in \mathcal{K}} f(x) \leq \frac{3G^2}{2H}e^{-\frac{H}{\beta}T}\left(1 + \frac{H}{\beta}T\right) . $$

**Intuition:** For strongly-convex objectives the appropriate GD algorithm utilizes two very extreme learning rates of $\eta_t \propto 1/t$ vs. $\eta_t = 1/\beta$ for the general/smooth settings respectively. A possible explanation to the universality of SCAdaNGD$_2$ is that it implicitly interpolate between these rates. Indeed the update rule of our algorithm can be written as follows, $x_{t+1} = x_t - \frac{1}{H}\frac{\|g_t\|^{-2}}{\sum_{\tau=1}^{t}\|g_\tau\|^{-2}}g_t$. Thus, ignoring the hindsight weighting, SCAdaNGD$_2$ is equivalent to GD with an adaptive learning rate $\tilde{\eta}_t := \|g_t\|^{-2}/H\sum_{\tau=1}^{t}\|g_\tau\|^{-2}$. Now, when all gradient norms are of the same magnitude, then $\tilde{\eta}_t \propto 1/t$, which boils down to the standard GD for strongly-convex objectives. Conversely, assume that the gradients are exponentially decaying, i.e., that $\|g_t\| \propto q^t$ for some $q < 1$. In this case $\tilde{\eta}_t$ is approximately constant. We believe that the latter applies for strongly-convex & smooth case.

## 4 Adaptive NGD for Stochastic Optimization

Here we show that using data-dependent minibatch sizes, we can adapt our (SC-)AdaNGD$_2$ algorithms (Algs. 2, 3 with $k = 2$) to the stochastic setting, and achieve the well know convergence rates for the convex/strongly-convex settings. Next we introduce the stochastic optimization setting, and then we present and discuss our Lazy SGD algorithm.

**Setup:** We consider the problem of minimizing a convex/strongly-convex function $f : \mathcal{K} \mapsto \mathbb{R}$, where $\mathcal{K} \in \mathbb{R}^d$ is a convex set. We assume that optimization lasts for $T$ rounds; on each round

---

**Algorithm 4** Lazy Stochastic Gradient Descent (LazySGD)

---

**Input**: #Oracle Queries $T$, $x_1 \in \mathbb{R}^d$, set $\mathcal{K}$, $\eta_0$, $p$
Set: $t = 0$, $s = 0$
**while** $t \leq T$ **do**
    Update: $s = s + 1$
    Set $\mathcal{G} = \text{GradOracle}(x_s)$, i.e., $\mathcal{G}$ generates i.i.d. noisy samples of $\nabla f(x_s)$
    Get: $(\tilde{g}_s, n_s) = \text{AE}(\mathcal{G}, T - t)$ % Adaptive Minibatch
    Update: $t = t + n_s$
    Calculate: $\hat{g}_s = n_s \tilde{g}_s$
    Set: $\eta_s = \eta_0 / t^p$
    Update: $x_{s+1} = \Pi_{\mathcal{K}}(x_s - \eta_s \hat{g}_s)$
**end while**
**Return**: $\bar{x}_T = \sum_{i=1}^{s} \frac{n_i}{T} x_i$ . (Note that $\sum_{i=1}^{s} n_i = T$)

---

---

**Algorithm 5** Adaptive Estimate (AE)

---

**Input**: random vectors generator $\mathcal{G}$, sample budget $T_{\max}$, sample factor $m_0$
Set: $i = 0, N = 0, \tilde{g}_0 = 0$
**while** $N < T_{\max}$ **do**
    Take $\tau_i = \min\{2^i, T_{\max} - N\}$ samples from $\mathcal{G}$
    Set $N \leftarrow N + \tau_i$
    Update: $\tilde{g}_N \leftarrow$ Average of N samples received so far from $\mathcal{G}$
    **If** $\|\tilde{g}_N\| > 3m_0/\sqrt{N}$ **then return** $(\tilde{g}_N, N)$
    Update $i \leftarrow i + 1$
**end while**
**Return**: $(\tilde{g}_N, N)$

---

$t = 1, \dots, T$, we may query a point $x_t \in \mathcal{K}$, and receive a *feedback*. After the last round, we choose $\bar{x}_T \in \mathcal{K}$, and our performance measure is the expected excess loss, defined as,

$$\mathbf{E}[f(\bar{x}_T)] - \min_{x \in \mathcal{K}} f(x) .$$

Here we assume that our feedback is a first order noisy oracle $\mathcal{G} : \mathcal{K} \mapsto \mathbb{R}^d$ such that upon querying $\mathcal{G}$ with a point $x_t \in \mathcal{K}$, we receive a bounded and unbiased gradient estimate, $\mathcal{G}(x_t)$, such $\mathbf{E}[\mathcal{G}(x_t)|x_t] = \nabla f(x_t)$; $\|\mathcal{G}(x_t)\| \leq G$. We also assume that the that the internal coin tosses (randomizations) of the oracle are independent. It is well known that variants of Stochastic Gradient Descent (SGD) are ensured to output an estimate $\bar{x}_T$ such that the excess loss is bounded by $O(1/\sqrt{T})/O(1/T)$ for the setups of convex/strongly-convex stochastic optimization, [20], [21].
 **Notation:** In this section we make a clear distinction between the number of queries to the gradient oracle, denoted henceforth by $T$; and between the number of iterations in the algorithm, denoted henceforth by $S$. We care about the dependence of the excess loss in $T$.

### 4.1 Lazy Stochastic Gradient Descent

**Data Dependent Minibatch sizes:** The Lazy SGD (Alg. 4) algorithm that we present in this section, uses a minibatch size that changes in between query points. Given a query point $x_s$, Lazy SGD invokes the noisy gradient oracle $\tilde{O}(1/\|g_s\|^2)$ times, where $g_s := \nabla f(x_s)$ [3]. Thus, in contrast to SGD which utilizes a fixed number of oracle calls per query point, our algorithm tends to stall in points with smaller gradients, hence the name Lazy SGD.

Here we give some intuition regarding our adaptive minibatch size rule: Consider the stochastic optimization setting. However, imagine that instead of the noisy gradient oracle $\mathcal{G}$, we may access an improved (imaginary) oracle which provides us with unbiased estimates, $\tilde{g}(x)$, that are accurate up to some *multiplicative factor*, e.g., $\mathbf{E}[\tilde{g}(x)|x] = \nabla f(x)$, and $\frac{1}{2}\|\nabla f(x)\| \leq \|\tilde{g}(x)\| \leq 2\|\nabla f(x)\|$ . Then intuitively we could have used these estimates instead of the exact normalized gradients inside our (SC-)AdaNGD$_2$ algorithms (Algs. 2, 3 with $k = 2$), and still get similar (in expectation) data

dependent bounds. Quite nicely, we may use our original noisy oracle $\mathcal{G}$ to generate estimates from this imaginary oracle. This can be done by invoking $\mathcal{G}$ for $\tilde{O}(1/\|g_s\|^2)$ times at each query point. Using this minibatch rule, the total number of calls to $\mathcal{G}$ (along all iterations) is equal to $T = \sum_{s=1}^{S} 1/\|g_s\|^2$. Plugging this into the data dependent bounds of (SC-)AdaNGD$_2$ (Thms. 2.2, 3.2), we get the well known $\tilde{O}(1/\sqrt{T})/\tilde{O}(1/T)$ rates for the stochastic convex settings.

**The imaginary oracle:** The construction of the imaginary oracle from the original oracle appears in Algorithm 5 (AE procedure) . It receives as an input, $\mathcal{G}$, a generator of independent random vectors with an (unknown) expected value $g \in \mathbb{R}^d$. The algorithm outputs two variables: $N$ which is an estimate of $1/\|g\|^2$, and $\tilde{g}_N$ an average of $N$ random vectors from $\mathcal{G}$. Thus, it is natural to think of $N\tilde{g}_N$ as an estimate for $g/\|g\|^2$. Moreover, it can be shown that $E[N(\tilde{g}_N - g)] = 0$. Thus in a sense we receive an unbiased estimate. The guarantees of Algorithm 5 appear below,

**Lemma 4.1** (Informal). *Let $T_{\max} \geq 1, \delta \in (0,1)$. Suppose an oracle $\mathcal{G} : \mathcal{K} \mapsto \mathbb{R}^d$ that generates G-bounded i.i.d. random vectors with an (unknown) expected value $g \in \mathbb{R}^d$. Then w.p.$\geq 1 - \delta$, invoking AE (Algorithm 5), with $m_0 = \Theta(G \log(1/\delta))$, it is ensured that:*

$$N = \Theta(\min\{m_0/\|g\|^2, T_{\max}\}), \; and \; E[N(\tilde{g}_N - g)] = 0 \;.$$

**Lazy SGD:** Now, plugging the output of the AE algorithm into our offline algorithms (SC-)AdaNGD$_2$, we get their stochastic variants which appears in Algorithm 4 (Lazy SGD). This algorithm is equivalent to the offline version of (SC-)AdaNGD$_2$, with the difference that we use $n_s$ instead of $1/\|\nabla f(x_s)\|^2$ and $n_s\tilde{g}_s$ instead of $\nabla f(x_s)/\|\nabla f(x_s)\|^2$.

Let $T$ be a bound on the *total number of queries* to the the first order oracle $\mathcal{G}$, and $\delta$ be the confidence parameter used to set $m_0$ in the AE procedure. Next we present the guarantees of LazySGD,

**Lemma 4.2.** *Let $\delta = O(T^{-3/2})$; let $\mathcal{K}$ be a convex set with diameter D, and f be a convex function; and assume $\|\mathcal{G}(x)\| \leq G$ w.p.1. Then using LazySGD with $\eta_0 = D/\sqrt{2}G$, $p = 1/2$, ensures:*

$$\mathbf{E}[f(\bar{x}_T)] - \min_{x \in \mathcal{K}} f(x) \; \leq \; O\left(\frac{GD \log(T)}{\sqrt{T}}\right) \;.$$

**Lemma 4.3.** *Let $\delta = O(T^{-2})$, let $\mathcal{K}$ be a convex set, and f be an H-strongly-convex convex function; and assume $\|\mathcal{G}(x)\| \leq G$ w.p.1. Then using LazySGD with $\eta_0 = 1/H$, $p = 1$, ensures:*

$$\mathbf{E}[f(\bar{x}_T)] - \min_{x \in \mathcal{K}} f(x) \; \leq \; O\left(\frac{G^2 \log^2(T)}{HT}\right) \;.$$

Note that LazySGD uses minibatch sizes that are adapted to the magnitude of the gradients, and still maintains the optimal $O(1/\sqrt{T})/O(1/T)$ rates. In contrast using a fixed minibatch size $b$ for SGD might degrade the convergence rates, yielding $O(\sqrt{b}/\sqrt{T})/O(b/T)$ guarantees. This property of LazySGD may be beneficial when considering distributed computations (see [13]).

## 5 Discussion

We have presented a new approach based on a conversion scheme, which exhibits universality and new adaptive bounds in the offline convex optimization setting, and provides a principled approach towards minibatch size selection in the stochastic setting. Among the many questions that remain open is whether we can devise "accelerated" universal methods. Furthermore, our universality results only apply when the global minimum is inside the constraints. Thus, it is natural to seek for methods that ensure universality when this assumption is violated. Moreover, our algorithms depend on a parameter $k \in \mathbb{R}$, but only the cases where $k \in \{0, 1, 2\}$ are well understood. Investigating a wider spectrum of $k$ values is intriguing. Lastly, it is interesting to modify and test our methods in non-convex scenarios, especially in the context of deep-learning applications.

**Acknowledgments**

I would like to thank Elad Hazan and Shai Shalev-Shwartz for fruitful discussions during the early stages of this work.

This work was supported by the ETH Zürich Postdoctoral Fellowship and Marie Curie Actions for People COFUND program.

## Footnotes

[1]For concreteness we concentrate in this work on converting AdaGrad, [1]. Note that our conversion scheme applies more widely to other adaptive online methods.

[2]A degradation by a $\sqrt{b}$ factor in the general case and by a $b$ factor in the strongly-convex case.

[3]Note that the gradient norm, $\|g_s\|$, is unknown to the algorithm. Nevertheless it is estimated on the fly.

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
