[Supplementary Material]

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

## A  Extensions

**Acceleration:** The catalyst approach, [22], enables to take any first order method that ensures linear convergence rates in the strongly-convex and smooth case and transform it into an accelerated method obtaining $O(\exp(-\sqrt{\gamma}T))$ rate in the strongly-convex and smooth case, and $O(1/T^2)$ rate in the smooth case. In particular, this acceleration applies to our SC-AdaNGD$_2$ Algorithm. Unfortunately, the catalyst approach requires the smoothness parameter, and the resulting accelerated SC-AdaNGD$_2$ is no longer universal.

**Other adaptive online schemes:** The adaptive methods that we have presented so far lean on AdaGrad (Alg. 1). Nevertheless, we may base our methods on other online algorithms with adaptive regret guarantees, and obtain convergence rates of the form,

$$f(\bar{x}_T) - \min_{x \in \mathcal{K}} f(x) \leq \frac{\mathcal{R}^{\mathcal{A}}\left(g_1/\|g_1\|^k, \ldots, g_T/\|g_T\|^k\right)}{\sum_{t=1}^{T}\|g_t\|^k} \,,$$

where $\mathcal{R}^{\mathcal{A}}(\theta_1, \ldots, \theta_T)$ is the regret bound of algorithm $\mathcal{A}$ with respect to the linear loss sequence $\{\theta_t^\top x\}_{t=1}^{T}$. For example we can use the very popular version of AdaGrad, which employs a separate learning rate to different directions. Also noteworthy is the Multiplicative Weights (MW) online algorithm, which over the simplex ensures a regret bound of the form (see [23], [24]),

$$\mathcal{R}^{MW} \leq \sqrt{\sum_{t=1}^{T}\|g_t\|_\infty^2 \log(d)} \,.$$

Using AdaNGD$_k$ with the appropriate modifications: AdaGrad$\leftrightarrow$MW, and $\ell_2 \leftrightarrow \ell_\infty$, yields similar adaptive guarantees as in Theorems 2.1, 2.2, with the difference that, $D \leftrightarrow \log d$, and $\ell_2 \leftrightarrow \ell_\infty$.

## B  Experiments

As a preliminary experimental investigation we compare our SC-AdaNGD$_k$ to GD accelerated-GD, and line-search for two strongly-convex objectives[4]. Concretely, we compare the above methods for the following quadratic (smooth) minimization problem,

$$\min_{x \in \mathbb{R}^d} R(x) := \frac{1}{2} \sum_{i=1}^{d} i \cdot x_i^2 \,.$$

,and also for the following non-smooth problem,

$$\min_{\|x\| \leq 1} F(x) := \frac{1}{2} \sum_{i=1}^{d} i \cdot x_i^2 + \|x\|_1 \,.$$

where $x_i$ is the $i$'th component of $x$, and $\|x\|_1$ is the $\ell_1$ norm. Note that both $R$ and $F$ are 1-strongly-convex, however $R$ is $d$-smooth while $F$ is non-smooth. Also, for both $R$ and $F$ the unique global minimum is in $x = 0$. We initialize all of the methods at the same random point, and take $d = 100$.

The results are depicted in Fig. 1. In Fig. 1(a) we present our results for the smooth quadratic objective $R$. We compare three SC-AdaNGD$_k$ variants $k \in \{1, 1.1, 2\}$, to GD which uses a constant learning rate $\eta_t = 1/\beta$ (recall $\beta = d = 100$), and to Nesterov's accelerated method. While this is not surprising that the latter demonstrates the best performance, it is surprising that all SC-AdaNGD$_k$ variants are performing better than GD/lines-search, and the $k = 1.1$ variant substantially outperforms GD. Also, in contrast to GD, SC-AdaNGD$_k$ are not descent methods, in the sense that the losses are not necessarily monotonically decreasing from one iteration to another.

Fig. 1(b) shows the results for the non-smooth objective $F$, where we compare two SC-AdaNGD$_k$ variants $k \in \{1, 2\}$, with two variants of GD, **(i)** const learning rate $\eta_t = 1/\beta$, and **(ii)** decaying

Figure 1: SC-AdaNGD$_k$ compared to GD, accelerated-GD and line-search. Left: strongly-convex and smooth objective, $R(\cdot)$. Middle: strongly-convex and non-smooth objective, $F(\cdot)$. Right: iterates of these methods for a 2D quadratic objective, $Z(\cdot)$.

Figure 2: Robustness experiments comparing SC-AdaNGD$_k$ with GD and accelerated-GD for the strongly-convex and smooth objective, $R(\cdot)$. Gradient oracle is perturbed with $\propto 10^{-6}$ noise magnitude.

learning rate $\eta_t = 1/Ht$. We have also compared to accelerated-GD and found its performance to be similar to GD-const (and therefore omitted). As can be seen, GD with a constant learning rate is doing very poorly, SC-AdaNGD$_2$ demonstrates the best performance, and GD-SC (decay) lags behind only by little. Note that for GD-SC (decay) we present results for a moving average over the GD iterates (which improve its performance).

The universality of SC-AdaNGD$_k$ for $k \in \{1, 2\}$ is clearly evident from Figures 1(a) ,1(b). In order to learn more about the character of SC-AdaNGD, we have applied the above methods to a simple 2D quadratic objective,

$$Z(x) = x_1^2 + 10x_2^2 .$$

The progress (iterates) of these methods is presented in Fig. 1(c). It can be seen that GD and accelerated-GD converge quickly to the $x_1$ axis and progress along it towards $(0, 0)$. Conversely, SC-AdaNGD methods progress diagonally, however take larger steps in the $x_1$ directions compared to GD and accelerated-GD.

**Robustness:** We have also examined the robustness of SC-AdaNGD compared to GD, accelerated-GD and line-search. We applied these methods to the quadratic objective $R$, however instead of the exact gradients we provided them with a slightly noisy and (unbiased) gradient feedback. The results when using noise perturbation magnitude of $10^{-6}$ appear in Fig. 2. This behaviour persisted when we employed other noise magnitudes.

**Stochastic setting:** We made a few experiments in the stochastic setting. While examining LazySGD, we have found out that using the $n_s$ output of the AE procedure (Alg. 5) is a too crude estimate for $1/\|g_s\|^2$ (due to the doubling procedure), which lead to unsatisfactory performance. Instead, we found that using $1/\|\tilde{g}_s\|^2$ is a much better approximation, that works very well in practice.

An initial experimental study on several simple stochastic problems shows that LazySGD (with the above modification) compares with minibatch SGD, for various values of minibatch sizes. A more elaborate examination of LazySGD is left for future work.

## C   Proofs for Section 2 (AdaNGD)

### C.1   Proof of Theorem 1.1 (AdaGrad)

*Proof.* Let $x \in \mathcal{K}$ and Consider the update rule $x_{t+1} = \Pi_{\mathcal{K}}(x_t - \eta_t g_t)$. We can write:

$$\|x_{t+1} - x\|^2 \le \|x_t - x\|^2 - 2\eta_t g_t(x_t - x) + \eta_t^2 \|g_t\|^2$$

Re-arranging the above we get:

$$g_t(x_t - x) \le \frac{1}{2\eta_t} \left( \|x_t - x\|^2 - \|x_{t+1} - x\|^2 \right) + \frac{\eta_t}{2} \|g_t\|^2 .$$

Combined with the convexity of $f_t$ and summing over all rounds we conclude that $\forall x \in \mathcal{K}$,

$$\sum_{t=1}^{T} f_t(x_t) - \sum_{t=1}^{T} f_t(x) \le \sum_{t=1}^{T} \frac{\|x_t - x\|^2}{2} \left( \frac{1}{\eta_t} - \frac{1}{\eta_{t-1}} \right) + \sum_{t=1}^{T} \frac{\eta_t}{2} \|g_t\|^2$$

$$\le \frac{D^2}{2} \sum_{t=1}^{T} \left( \frac{1}{\eta_t} - \frac{1}{\eta_{t-1}} \right) + \frac{D}{2\sqrt{2}} \sum_{t=1}^{T} \frac{\|g_t\|^2}{\sqrt{\sum_{\tau=1}^{t} \|g_\tau\|^2}}$$

$$\le \frac{D}{2} \sqrt{2 \sum_{t=1}^{T} \|g_t\|^2} + \frac{D}{\sqrt{2}} \sqrt{\sum_{t=1}^{T} \|g_t\|^2}$$

$$= \sqrt{2D^2 \sum_{t=1}^{T} \|g_t\|^2}$$

here in the first inequality we denote $\eta_0 = \infty$, the second inequality uses diam$\mathcal{K} = D$ and $\eta_t \le \eta_{t-1}$, the third inequality uses the following lemma from [6]:

**Lemma C.1.** *For any non-negative numbers $a_1, \ldots, a_n$ the following holds:*

$$\sum_{i=1}^{n} \frac{a_i}{\sqrt{\sum_{j=1}^{i} a_j}} \le 2 \sqrt{\sum_{i=1}^{n} a_i}$$

□

### C.2   Proof of Lemma 2.1

*Proof.* Notice that AdaNGD$_k$ described in Algorithm 2, is equivalent to applying AdaGrad (Algorithm 1) to the following sequence of linear loss functions:

$$\left\{ \tilde{f}_t(x) := \frac{1}{\|g_t\|^k} g_t^\top x \right\}_{t=1}^{T} .$$

The regret bound of AdaGrad appearing in Theorem 1.1 implies the following for any $x \in \mathcal{K}$:

$$\sum_{t=1}^{T} \frac{1}{\|g_t\|^k} g_t^\top (x_t - x) \le \sqrt{2D^2 \sum_{t=1}^{T} 1/\|g_t\|^{2(k-1)}} . \tag{1}$$

Using the above bound together with Jensen's inequality, enables to bound the excess loss of $\text{AdaNGD}_k$:

$$
\begin{aligned}
f(\bar{x}_T) - f(x^*) \ &\leq \ \sum_{t=1}^{T} \frac{\|g_t\|^{-k}}{\sum_{\tau=1}^{T} \|g_\tau\|^{-k}} \left( f(x_t) - f(x^*) \right) \\
&\leq \ \sum_{t=1}^{T} \frac{\|g_t\|^{-k}}{\sum_{\tau=1}^{T} \|g_\tau\|^{-k}} g_t^\top (x_t - x^*) \\
&= \ \frac{1}{\sum_{\tau=1}^{T} \|g_\tau\|^{-k}} \sum_{t=1}^{T} \frac{1}{\|g_t\|^k} g_t^\top (x_t - x^*) \\
&\leq \ \frac{\sqrt{2D^2 \sum_{t=1}^{T} 1/\|g_t\|^{2(k-1)}}}{\sum_{\tau=1}^{T} 1/\|g_\tau\|^{k}} \ ,
\end{aligned}
$$

where the second line uses the gradient inequality. $\qquad\square$

### C.3 Proof of Theorem 2.1

*Proof.* The data dependent bound,

$$
f(\bar{x}_T) - f(x^*) \ \leq \ \frac{\sqrt{2D^2 T}}{\sum_{t=1}^{T} 1/\|g_t\|} \ , \tag{2}
$$

is a direct corollary of Lemma 2.1 with $k = 1$. Note that the above bound holds for both smooth/non-smooth cases. The general case bound holds directly by using $\|g_t\| \leq G$.

Next we focus on the second part of the theorem regarding the smooth case. We will first require the following lemma regarding smooth objectives,

**Lemma C.2.** *Let* $F : \mathbb{R}^d \mapsto \mathbb{R}$ *be a* $\beta$*-smooth function, and let* $x^* = \arg\min_{x \in \mathbb{R}^d} F(x)$*, then,*

$$
\|\nabla F(x)\|^2 \ \leq \ 2\beta \left( F(x) - F(x^*) \right), \quad \forall x \in \mathbb{R}^d \ .
$$

The above lemma enables to upper bound sum of gradient norms in the query points of $\text{AdaNGD}_1$,

$$
\begin{aligned}
\sum_{t=1}^{T} \|g_t\| \ &= \ \sum_{t=1}^{T} \frac{\|g_t\|^2}{\|g_t\|} \\
&\leq \ \sum_{t=1}^{T} \frac{2\beta}{\|g_t\|} \left( f(x_t) - f(x^*) \right) \\
&\leq \ \sum_{t=1}^{T} \frac{2\beta}{\|g_t\|} g_t^\top (x_t - x^*) \\
&= \ 2\beta \sum_{t=1}^{T} \hat{g}_t^\top (x_t - x^*) \\
&\leq \ 2\sqrt{2}\beta D \sqrt{T} \ , \tag{3}
\end{aligned}
$$

where the last line follows by the regret guarantee of AdaGrad for the following sequence (see Equation (1)),

$$
\left\{ \tilde{f}_t(x) := \frac{1}{\|g_t\|} g_t^\top x \right\}_{t=1}^{T} \ .
$$

The second line is a consequence of Lemma C.2 regarding smooth objectives. Now utilizing the convexity of the function $H(z) = 1/z$ for $z > 0$, and applying Equation (3), we may bound the sum

of inverse gradients:

$$
\sum_{\tau=1}^{T} \frac{1}{\|g_\tau\|} = T \frac{1}{T} \sum_{\tau=1}^{T} \frac{1}{\|g_\tau\|} \geq T \frac{1}{\frac{1}{T} \sum_{\tau=1}^{T} \|g_\tau\|}
$$

$$
\geq T \frac{1}{2\sqrt{2}\beta D/\sqrt{T}} .
$$

Rearranging the latter equation, and using Equation (2) concludes the proof,

$$
f(\bar{x}_T) - \min_{x \in \mathcal{K}} f(x) \leq \frac{D\sqrt{2T}}{\sum_{\tau=1}^{T} 1/\|g_\tau\|} \leq \frac{4\beta D^2}{T} .
$$

$\square$

## C.4  Proof of Theorem 2.2

*Proof.* The data dependent bound,

$$
f(\bar{x}_T) - f(x^*) \leq \frac{\sqrt{2D^2}}{\sqrt{\sum_{t=1}^{T} 1/\|g_t\|^2}} , \tag{4}
$$

is a direct corollary of Lemma 2.1 with $k = 2$. Note that the above bound holds for both smooth/non-smooth cases. The general case bound holds directly by using $\|g_t\| \leq G$.

We will now focus on the second part of the theorem regarding the smooth case. Let us lower bound $\sum_{t=1}^{T} 1/\|g_t\|^2$ for AdaNGD$_2$:

$$
T = \sum_{t=1}^{T} \frac{\|g_t\|^2}{\|g_t\|^2}
$$

$$
\leq \sum_{t=1}^{T} \frac{2\beta}{\|g_t\|^2} \left( f(x_t) - f(x^*) \right)
$$

$$
\leq \sum_{t=1}^{T} \frac{2\beta}{\|g_t\|^2} g_t^\top (x_t - x^*)
$$

$$
= 2\beta \sum_{t=1}^{T} \left( \tilde{f}_t(x_t) - \tilde{f}_t(x^*) \right)
$$

$$
\leq 2\sqrt{2}\beta D \sqrt{\sum_{t=1}^{T} \frac{1}{\|g_t\|^2}} , \tag{5}
$$

where the last line follows by the regret guarantee of AdaGrad for the following sequence (see Equation (1)),

$$
\left\{ \tilde{f}_t(x) = \frac{1}{\|g_t\|^2} g_t^\top x \right\}_{t=1}^{T} .
$$

The second line is a consequence of Lemma C.2. Combining Equation (5) together with Equation (4) concludes the proof. $\square$

## C.5  Proof of Lemma C.2

*Proof.* The $\beta$ smoothness of $F$ means the following to hold $\forall x, u \in \mathbb{R}^d$,

$$
F(x + u) \leq F(x) + \nabla F(x)^\top u + \frac{\beta}{2} \|u\|^2 .
$$

Taking $u = -\frac{1}{\beta} \nabla F(x)$ we get,

$$
F(x + u) \leq F(x) - \frac{1}{\beta} \|\nabla F(x)\|^2 + \frac{1}{2\beta} \|\nabla F(x)\|^2 .
$$

---

**Algorithm 6** Strongly-Convex Adaptive Gradient Descent (SC-AdaGrad)

---

**Input**: #Iterations $T$, $x_1 \in \mathbb{R}^d$, set $\mathcal{K}$
Set: $Q_0 = 0$
**for** $t = 1 \dots T$ **do**
    Calculate: $g_t = \nabla f_t(x_t)$
    Let: $H_t$ be the strong-convexity parameter of $f_t(\cdot)$
    Update:
$$Q_t = Q_{t-1} + H_t$$

    Set $\eta_t = 1/Q_t$
    Update:
$$x_{t+1} = \Pi_{\mathcal{K}}(x_t - \eta_t g_t)$$

**end for**

---

Thus:

$$
\begin{aligned}
\|\nabla F(x)\| &\leq \sqrt{2\beta\big(F(x) - F(x+u)\big)} \\
&\leq \sqrt{2\beta\big(F(x) - F(x^*)\big)} ,
\end{aligned}
$$

where in the last inequality we used $F(x^*) \leq F(x+u)$ which holds since $x^*$ is the *global* minimum.

$\square$

# D    Proofs for Section 3 (SC-AdaNGD)

## D.1    Proof of Lemma 3.1

*Proof.* We will require the following extension of Theorem 1 from [21]. Its proof is provided in Section D.4.

**Lemma D.1** (SC-AdaGrad, Alg 6). *Assume that we receive a sequence of convex loss functions $f_t : \mathcal{K} \mapsto \mathbb{R}$, $t \in [T]$, and suppose that each function $f_t$ is $H_t$-strongly-convex. Using the update rule $x_{t+1} = \Pi_{\mathcal{K}}(x_t - \eta_t g_t)$ where $g_t = \nabla f_t(x_t)$ and $\eta_t = (\sum_{\tau=1}^t H_\tau)^{-1}$ yields the following regret bound:*

$$
\sum_{t=1}^T f_t(x_t) - \sum_{t=1}^T f_t(x) \leq \frac{1}{2} \sum_{t=1}^T \eta_t \|g_t\|^2 .
$$

We are now ready to go on with the proof. Note that SC-AdaNGD$_k$ depicted in Algorithm 3 is equivalent to performing SC-AdaGrad updates $x_{t+1} = \Pi_{\mathcal{K}}(x_t - \eta_t \nabla \tilde{f}_t(x_t))$ over the following loss sequence:

$$
\left\{ \tilde{f}_t(x) = \frac{1}{\|g_t\|^k} g_t^\top x + \frac{H}{2\|g_t\|^k} \|x - x_t\|^2 \right\}_{t=1}^T
$$

where $g_t = \nabla f_t(x_t)$. Note that each $\tilde{f}_t(x)$ is $\frac{H}{\|g_t\|^k}$-strongly-convex, and that the learning rate is inversely proportional to the cumulative sum of strong-convexities. Thus Lemma D.1 implies the following to hold for any $x \in \mathcal{K}$:

$$
\sum_{t=1}^T \tilde{f}_t(x_t) - \sum_{t=1}^T \tilde{f}_t(x) \leq \frac{1}{2H} \sum_{t=1}^T \frac{\|g_t\|^{-2(k-1)}}{\sum_{\tau=1}^t \|g_\tau\|^{-k}} .
$$

Combining the latter bound with the definition of $\bar{x}_T$, and applying Jensen's inequality we conclude:

$$
\begin{aligned}
f(\bar{x}_T) - f(x^*) &\leq \sum_{t=1}^{T} \frac{\|g_t\|^{-k}}{\sum_{\tau=1}^{T} \|g_\tau\|^{-k}} \left( f(x_t) - f(x^*) \right) \\
&\leq \frac{1}{\sum_{t=1}^{T} \|g_t\|^{-k}} \sum_{t=1}^{T} \|g_t\|^{-k} \left( g_t^\top (x_t - x^*) - \frac{H}{2} \|x_t - x^*\|^2 \right) \\
&= \frac{1}{\sum_{t=1}^{T} \|g_t\|^{-k}} \sum_{t=1}^{T} \left( \tilde{f}_t(x_t) - \tilde{f}_t(x^*) \right) \\
&\leq \frac{1}{2H \sum_{t=1}^{T} \|g_t\|^{-k}} \sum_{t=1}^{T} \frac{\|g_t\|^{-2(k-1)}}{\sum_{\tau=1}^{t} \|g_\tau\|^{-k}} ,
\end{aligned}
$$

where we used the $H$-strong-convexity of $f$ in the second line. $\qquad\square$

## D.2  Proof of Theorem 3.1

*Proof.* We will require the following lemma, its proof is provided in Section D.5.

**Lemma D.2.** *For any non-negative real numbers $a_1, \ldots, a_n \geq 1$,*

$$
\sum_{i=1}^{n} \frac{a_i}{\sum_{j=1}^{i} a_j} \leq 1 + \log \left( \sum_{i=1}^{n} a_i \right) .
$$

Combining the above lemma together with Lemma 3.1 and using $k = 1$, we obtain,

$$
\begin{aligned}
f(\bar{x}_T) - f(x^*) &\leq \frac{1}{2H \sum_{t=1}^{T} \|g_t\|^{-1}} \sum_{t=1}^{T} \frac{1}{\sum_{\tau=1}^{t} \|g_\tau\|^{-1}} \\
&\leq \frac{1}{2H \sum_{t=1}^{T} \|g_t\|^{-1}} \sum_{t=1}^{T} \frac{G\|g_t\|^{-1}}{\sum_{\tau=1}^{t} \|g_\tau\|^{-1}} \\
&\leq \frac{G}{2H \sum_{t=1}^{T} \|g_t\|^{-1}} \sum_{t=1}^{T} \frac{G\|g_t\|^{-1}}{\sum_{\tau=1}^{t} G\|g_\tau\|^{-1}} \\
&\leq \frac{G}{2H \sum_{t=1}^{T} \|g_t\|^{-1}} \left( 1 + \log \left( \sum_{t=1}^{T} \frac{G}{\|g_t\|} \right) \right)
\end{aligned}
$$

where the second line uses $\|g_t\| \leq G$, and the last line uses Lemma D.2. Note that the above bound holds for both smooth/non-smooth cases.

We now turn to prove the second part of the theorem regarding the smooth case. First let us bound the sum of gradient norms in the query points of SC-AdaNGD$_1$:

$$
\begin{aligned}
\sum_{t=1}^{T} \|g_t\| &= \sum_{t=1}^{T} \frac{\|g_t\|^2}{\|g_t\|} \\
&\leq \sum_{t=1}^{T} \frac{2\beta}{\|g_t\|} \left( f(x_t) - f(x^*) \right) \\
&\leq \sum_{t=1}^{T} \frac{2\beta}{\|g_t\|} \left( g_t^\top (x_t - x^*) - \frac{H}{2} \|x_t - x^*\|^2 \right) \\
&= 2\beta \sum_{t=1}^{T} \left( \tilde{f}_t(x_t) - \tilde{f}_t(x^*) \right) \\
&\leq \frac{\beta}{H} \sum_{t=1}^{T} \frac{1}{\sum_{\tau=1}^{t} \|g_\tau\|^{-1}} \\
&\leq \frac{\beta}{H} G \left( 1 + \log \left( \sum_{t=1}^{T} \frac{G}{\|g_t\|} \right) \right) ,
\end{aligned}
$$

where the second line uses Lemma C.2, the third line uses the strong-convexity of $f$, the fourth line uses the regret bound of the SC-AdaGrad algorithm over the following sequence (see Equation (**??**)),

$$
\left\{ \tilde{f}_t(x) = \frac{1}{\|g_t\|} g_t^\top x + \frac{H}{2\|g_t\|} \|x - x_t\|^2 \right\}_{t=1}^{T} ,
$$

and the last line uses Lemma D.2. Combining the convexity of the function $H(z) = 1/z$ for $z > 0$, together with the above inequality, we may bound the sum of inverse gradient norms,

$$
\begin{aligned}
\sum_{\tau=1}^{T} \frac{1}{\|g_\tau\|} &= T \frac{1}{T} \sum_{\tau=1}^{T} \frac{1}{\|g_\tau\|} \geq T \frac{1}{\frac{1}{T} \sum_{\tau=1}^{T} \|g_\tau\|} \\
&\geq T^2 \frac{1}{(\beta/H) G \left( 1 + \log \left( \sum_{t=1}^{T} \frac{G}{\|g_t\|} \right) \right)} .
\end{aligned}
$$

Rearranging the latter equation, and using the data dependent bound for SC-AdaNGD$_1$ concludes the proof,

$$
f(\bar{x}_T) - \min_{x \in \mathcal{K}} f(x) \leq \frac{(\beta/H) G^2 (1 + \log T)^2}{H T^2} .
$$

$\square$

## D.3 Proof of Theorem 3.2

*Proof.* The data dependent bound,

$$
f(\bar{x}_T) - \min_{x \in \mathcal{K}} f(x) \leq \frac{1 + \log(G^2 \sum_{t=1}^{T} \|g_t\|^{-2})}{2H \sum_{t=1}^{T} \|g_t\|^{-2}} \tag{6}
$$

is a direct corollary of Lemma 3.1 with $k = 2$, combined with Lemma D.2. Note that the above bound holds for both smooth/non-smooth cases.

We now turn to prove the second part of the theorem regarding the smooth case. Let us lower bound $\sum_{t=1}^{T} 1/\|g_t\|^2$, for SC-AdaNGD$_2$:

$$
\begin{aligned}
T &= \sum_{t=1}^{T} \frac{\|g_t\|^2}{\|g_t\|^2} \\
&\leq \sum_{t=1}^{T} \frac{2\beta}{\|g_t\|^2} \left( f(x_t) - f(x^*) \right) \\
&\leq \sum_{t=1}^{T} \frac{2\beta}{\|g_t\|^2} \left( g_t^\top (x_t - x^*) - \frac{H}{2} \|x_t - x^*\|^2 \right) \\
&= 2\beta \sum_{t=1}^{T} \left( \tilde{f}_t(x_t) - \tilde{f}_t(x^*) \right) \\
&\leq \frac{\beta}{H} \sum_{t=1}^{T} \frac{\|g_t\|^{-2}}{\sum_{\tau=1}^{t} \|g_\tau\|^{-2}} \\
&\leq \frac{\beta}{H} \left( 1 + \log(G^2 \sum_{t=1}^{T} \|g_t\|^{-2}) \right) ,
\end{aligned}
\tag{7}
$$

where the second line uses Lemma C.2, the third line uses the strong-convexity of $f$, the fifth line uses the regret bound of the SC-AdaGrad algorithm for the following sequence (see Equation (**??**)),

$$
\left\{ \tilde{f}_t(x) = \frac{1}{\|g_t\|^2} g_t^\top x + \frac{H}{2\|g_t\|^2} \|x - x_t\|^2 \right\}_{t=1}^{T} ,
$$

and the last line uses Lemma D.2. Now Equation (7) implies,

$$
G^2 \sum_{t=1}^{T} \|g_t\|^{-2} \geq \frac{1}{3} e^{\frac{H}{\beta} T} .
\tag{8}
$$

Now let $z \in \mathbb{R}$ and note that the function $A(z) := \frac{1+\log(z)}{z}$ is monotonically decreasing for $z \geq 1$. Let $z = G^2 \sum_t \|g_t\|^{-2}$ and assume $\frac{1}{3} e^{\frac{H}{\beta} T} \geq 1$; combining this with Equation (6),(8), concludes the proof. Note that the case where $\frac{1}{3} e^{\frac{H}{\beta} T} \leq 1$ is not too interesting.

$\square$

## D.4 Proof of Lemma D.1

*Proof.* Let $x \in \mathcal{K}$ and Consider the update rule $x_{t+1} = \Pi_{\mathcal{K}}(x_t - \eta_t g_t)$. We can write:

$$
\|x_{t+1} - x\|^2 \leq \|x_t - x\|^2 - 2\eta_t g_t (x_t - x) + \eta_t^2 \|g_t\|^2 .
$$

Re-arranging the above we get:

$$
g_t(x_t - x) \leq \frac{1}{2\eta_t} \left( \|x_t - x\|^2 - \|x_{t+1} - x\|^2 \right) + \frac{\eta_t}{2} \|g_t\|^2 .
$$

Combining the above with the $H_t$-strong-convexity of $f_t$ and summing over all rounds we conclude that,

$$
\sum_{t=1}^{T} f_t(x_t) - \sum_{t=1}^{T} f_t(x) \leq \sum_{t=1}^{T} \frac{\|x_t - x\|^2}{2} \left( \frac{1}{\eta_t} - \frac{1}{\eta_{t-1}} - H_t \right) + \sum_{t=1}^{T} \frac{\eta_t}{2} \|g_t\|^2 ,
$$

where we denote $\eta_0 = \infty$. Recalling $\eta_t = (\sum_{\tau=1}^{t} H_\tau)^{-1}$, the lemma follows. $\square$

### D.5 Proof of Lemma D.2

*Proof.* We will prove the statement by induction over $n$. The base case $n = 1$ naturally holds. For the induction step, let us assume that the guarantee holds for $n - 1$, which implies that for any $a_1, \ldots, a_n \geq 1$,

$$\sum_{i=1}^{n} \frac{a_i}{\sum_{j=1}^{i} a_j} \leq 1 + \log\left(\sum_{i=1}^{n-1} a_i\right) + \frac{a_n}{\sum_{i=1}^{n} a_i} \ .$$

The above suggests that establishing following inequality concludes the proof,

$$1 + \log\left(\sum_{i=1}^{n-1} a_i\right) + \frac{a_n}{\sum_{i=1}^{n} a_i} \leq 1 + \log\left(\sum_{i=1}^{n} a_i\right) \ . \tag{9}$$

Using the notation $x = a_n / \sum_{i=1}^{n-1} a_i$, Equation (9) is equivalent to the following,

$$\log(x + 1) - \frac{x}{1 + x} \geq 0 \ .$$

However, it is immediate to validate that the function $M(x) = \log(x+1) - \frac{x}{1+x}$, is non-negative for any $x \geq 0$, which establishes the lemma. $\qquad\square$

## E  Proofs for Section 4.1 (Lazy SGD)

### E.1  Proof of Lemma 4.1

We first provide the exact statement rather than the informal one appearing in Lemma 4.1.

**Lemma E.1.** *Let* $T_{\max} \geq 1$. *Suppose an oracle* $\mathcal{G} : \mathcal{K} \mapsto \mathbb{R}^d$ *that generates i.i.d. random vectors with an (unknown) expected value* $g \in \mathbb{R}^d$. *Assume that w.p.* $1$ *the Euclidean norm of the sampled vectors is bounded by* $G$. *Then w.p.*$\geq 1 - \delta$, *invoking AE (Algorithm 5), with* $m_0 = 6G\left(1 + \sqrt{\log(\delta^{-1}(1 + \log_2 T_{\max}))}\right)$, *it is ensured that:*

$$\min\left\{m_0^2/\|g\|^2, T_{\max}\right\} \leq N \leq \min\left\{32m_0^2/\|g\|^2, T_{\max}\right\} \ . \textbf{(1)}$$

*Moreover, w.p.*$\geq 1 - \delta$, *the following holds for the output of the algorithm:*

$$\sqrt{N}\|\tilde{g}_N\| \leq 8m_0 \ . \textbf{(2)}$$

*and also,*

$$E[N(\tilde{g}_N - g)] = 0 \ . \textbf{(3)}$$

We will require the following Hoeffding type inequality regarding vector valued random variables, by [25] (see also [26])

**Theorem E.1.** *Suppose that* $X_1, X_2, \ldots, X_n \in \mathbb{R}^d$ *are i.i.d. random vectors, and that* $\forall i \in [n]; \|X_i\| \leq M$ *almost surely. Then w.p.*$\geq 1 - \delta$

$$\left\| \frac{1}{n} \sum_{i=1}^{n} X_i - \mathbf{E}[X_1] \right\| \leq \frac{6M}{\sqrt{n}} \left(1 + \sqrt{\log \delta^{-1}}\right) \ .$$

We are now ready to prove Lemma E.1.

*Proof of Lemma E.1.* Define $V = \left\{\{2^i - 1\}_{i=1}^{\log_2 T_{\max}}, T_{\max}\right\}$, and note that $N$ is a discrete random variable taking one of the $1 + \log_2 T_{\max}$ possible values among $V$. By Theorem E.1 combined with the union bound, it follows that w.p.$\geq 1 - \delta$, for every $n \in V$ we have $\|\tilde{g}_n - g\| \leq \frac{m_0}{\sqrt{n}}$. This means the following to hold:

$$\|\tilde{g}_n\| \leq \|g\| + \|\tilde{g}_n - g\| \leq \frac{2m_0}{\sqrt{n}}, \quad \forall n \in V \text{ such that } \|g\| \leq m_0/\sqrt{n} \tag{10}$$

Furthermore,

$$\|\tilde{g}_n\| \geq \|g\| - \|\tilde{g}_n - g\| \geq \frac{3m_0}{\sqrt{n}}, \quad \forall n \in V \text{ such that } \|g\| \geq 4m_0/\sqrt{n} \tag{11}$$

The above together with the stopping criteria of Algorithm 5 directly implies the first part of the lemma.

For the second part of the lemma, recall that $N$ is the total number of samples, and let $N_{\text{prev}}$ be the number of samples up to the iteration before stopping. Then necessarily, $N_{\text{prev}} \geq (N-1)/2$. Since the loop did not stop at the iteration before setting $N$, it follows that $\sqrt{N_{\text{prev}}}\|\tilde{g}_{N_{\text{prev}}}\| \leq 3m_0$ (i.e. the stopping criteria of the loop at the round prior to setting $N$ fails). Recalling that w.p.$\geq 1 - \delta$, for every $n \in V$ we have $\|\tilde{g}_n - g\| \leq \frac{m_0}{\sqrt{n}}$, and combining this with the above implies:

$$
\begin{aligned}
\sqrt{N}\|\tilde{g}_N\| &\leq \sqrt{N}\left(\|\tilde{g}_N - g\| + \|g - \tilde{g}_{N_{\text{prev}}}\|\right) + \sqrt{N}\|\tilde{g}_{N_{\text{prev}}}\| \\
&\leq \sqrt{N}\left(\frac{m_0}{\sqrt{N}} + \frac{m_0}{\sqrt{N_{\text{prev}}}}\right) + \sqrt{\frac{N}{N_{\text{prev}}}}\sqrt{N_{\text{prev}}}\|\tilde{g}_{N_{\text{prev}}}\| \\
&\leq m_0 + \sqrt{3}m_0 + \sqrt{3} \cdot 3m_0 \\
&\leq 8m_0
\end{aligned}
$$

Where we have used $N \leq 3\frac{N-1}{2} \leq 3N_{\text{prev}}$; which holds since $N_{\text{prev}} \geq (N-1)/2$ and also $N \geq 3$. The latter is ensured since for any $n \leq 3$ then $\|\tilde{g}_n\| \leq G < 3m_0/\sqrt{n}$.

For the third part of the lemma, it is easy to notice that for any fixed $n$ then $n(\tilde{g}_n - g)$ is a sum of $n$ i.i.d. random variables, and that $\mathbf{E}[n(\tilde{g}_n - g)] = 0$. Since $N$ is a bounded stopping time, Doob's optional stopping theorem [27] implies that $E[N(\tilde{g}_N - g)] = 0$. $\square$

### E.2 Proof of Lemma 4.2

*Proof.* Let $S$ be the total number of times that LazySGD invokes the AE procedure. We will first upper bound the expectation of following sum (weighted regret):

$$
\begin{aligned}
\sum_{s=1}^{S} n_s \left(f(x_s) - f(x^*)\right) &\leq \sum_{s=1}^{S} n_s g_s^\top (x_s - x^*) \\
&\leq \underbrace{\sum_{s=1}^{S} n_s \tilde{g}_s^\top (x_s - x^*)}_{(a)} + \underbrace{\sum_{s=1}^{S} n_s (g_s - \tilde{g}_s)^\top (x_s - x^*)}_{(b)} \tag{12}
\end{aligned}
$$

where we have used the gradient inequality. The proof goes on by bounding the expectation of terms $(a)$, $(b)$ appearing above.

**Bounding term (a):** Assume that LazySGD uses the AE procedure with some $\delta > 0$. Since LazySGD is equivalent to AdaNGD$_2$ with $\|g_s\|^2 \leftarrow n_s$ and $g_s \leftarrow n_s g_s$, then a similar analysis to AdaNGD$_2$ may show that this sum is bounded by $O(\sqrt{T})$. For completeness we provide the full analysis here. Consider the update rule of LazySGD: $x_{s+1} = \Pi_{\mathcal{K}}(x_s - \eta_s n_s \tilde{g}_s)$. We can write:

$$\|x_{s+1} - x^*\|^2 \leq \|x_s - x^*\|^2 - 2\eta_s n_s \tilde{g}_s^\top (x_t - x^*) + \eta_s^2 n_s^2 \|\tilde{g}_s\|^2$$

Re-arranging the above we get:

$$n_s \tilde{g}_s^\top (x_s - x^*) \leq \frac{1}{2\eta_s}(\|x_s - x^*\|^2 - \|x_{s+1} - x^*\|^2) + \frac{\eta_s}{2} n_s^2 \|\tilde{g}_s\|^2$$

Summing over all rounds we conclude that w.p.$\geq 1 - \delta T$:

$$
\begin{aligned}
\textbf{(a)} &= \sum_{s=1}^{S} n_s \tilde{g}_s^\top (x_s - x^*) \\
&\leq \sum_{s=1}^{S} \frac{\|x_s - x^*\|^2}{2}\left(\frac{1}{\eta_s} - \frac{1}{\eta_{s-1}}\right) + \sum_{s=1}^{S} \frac{\eta_s}{2} n_s^2 \|\tilde{g}_s\|^2 \\
&\leq \frac{D^2}{2} \sum_{s=1}^{S}\left(\frac{1}{\eta_s} - \frac{1}{\eta_{s-1}}\right) + 64 m_0^2 \sum_{s=1}^{S} \eta_s n_s \\
&\leq \frac{DG}{2}\sqrt{2T} + \frac{64 m_0^2 D}{G} \sum_{s=1}^{S} \frac{n_s}{\sqrt{\sum_{i=1}^{s} n_i}} \\
&= \frac{DG}{2}\sqrt{2T} + \frac{128 m_0^2 D}{G}\sqrt{\sum_{s=1}^{S} n_s} \\
&\leq O(GD\sqrt{T}\log(1/\delta)) \,.
\end{aligned}
$$

here in the first inequality we denote $\eta_0 = \infty$, the second inequality uses $n_s \|\tilde{g}_s\|^2 \leq 64 m_0^2$, which follows by Theorem E.1, and it also uses $\eta_s \leq \eta_{s-1}$; the fourth inequality uses Lemma C.1. We also make use of $\sum_{s=1}^{S} n_s = T$, and $1/\eta_s = \sqrt{\sum_{i=1}^{s} n_i}$.

Since **(a)** is bounded by $2GDT$, then taking $\delta = 1/T^{3/2}$ ensures that,

$$
\mathbf{E}[\textbf{(a)}] \;\leq\; O(GD\sqrt{T}\log(T)) \,. \tag{13}
$$

**Bounding term (b):**   Here we show that $\mathbf{E}[\textbf{(b)}] = 0$. Without loss of generality we will make the following two assumptions which do not affect the output of LazySGD:

- We assume that LazySGD invokes the AE procedure exactly $T$ times. Note that in practice the algorithm invokes the AE procedure $S$ times, where $S \leq T$ is a random variable, after which $T - t = 0$. Nevertheless calling AE for any $s \in \{S+1, \ldots, T\}$ yields $\tilde{g}_s = 0, n_s = 0$, which does not affect the output of LazySGD.

- We assume that at each time $s \in [T]$ that LazySGD calls the AE procedure, it samples exactly $T$ times from GradOracle$(x_s)$. We denote these samples by $\{\tilde{g}_s^{(i)}\}_{i=1}^{T}$. Nevertheless the output of the procedure only uses the first $n_s$ samples, where $n_s$ is set according to the AE procedure. Thus the remaining $T - n_s$ samples do not affect the output of AE and LazySGD. Note that $\forall s \in [T], n_s \leq T - t \leq T$,

Thus, for any $s \in [T]$ let $\{\tilde{g}_s^{(i)}\}_{i=1}^{T}$ be the samples drawn from the noisy first order oracle GradOracle$(x_s)$ during the $s$'th call to AE at this iteration. This implies that $n_s \tilde{g}_s = \sum_{i=1}^{n_s} \tilde{g}_s^{(i)}$. Term $(b)$ can be therefore written as follows:

$$
\textbf{(b)} = \sum_{s=1}^{T} n_s(g_s - \tilde{g}_s)^\top(x_s - x^*) = \sum_{s=1}^{T}\sum_{i=1}^{n_s}(g_s - \tilde{g}_s^{(i)})^\top(x_s - x^*)
$$

Given $s \in [T]$ define the following filtration:

$$
\begin{aligned}
\mathcal{F}_0^{(s)} &= \sigma\text{-field}\,\{x_s, t\} \\
\mathcal{F}_j^{(s)} &= \sigma\text{-field}\left\{x_s, t, g_s^{(1)}, \ldots, g_s^{(j)}\right\}, \quad \forall j \in [T]
\end{aligned}
$$

Also define the following sequence $\{B_j^{(s)}\}_{j=0}^{T}$:

$$
B_0^{(s)} = 0, \qquad B_j^{(s)} = \sum_{i=1}^{j}(g_s - \tilde{g}_s^{(i)})^\top(x_s - x^*), \quad \forall j \in [T]
$$

Since $\mathbf{E}[\tilde{g}_s^{(i)}|x_s] = g_s$, $\forall i, s \in [T]$, then it immediately follows that $\{B_j^{(s)}\}_{t=0}^{T}$ is a martingale with respect to the above filtration. Also it is immediate to see that $n_s$ is a bounded stopping time with respect to the above filtration. Thus, Doob's optional stopping theorem (see [27]) implies that

$$\mathbf{E}[B_{n_s}^{(s)}|\mathcal{F}_0] \;=\; \mathbf{E}\left[\sum_{i=1}^{n_s}(g_s - \tilde{g}_s^{(i)})^{\top}(x_s - x^*)|\mathcal{F}_0\right] \;=\; 0\,.$$

which directly implies,

$$\mathbf{E}[\mathbf{(b)}] \;=\; \mathbf{E}\left[\sum_{s=1}^{T} B_{n_s}^{(s)}\right] \;=\; 0\,.$$

Using Jensen's inequality and combining the above with Equations (12), (13), establishes the lemma:

$$\mathbf{E}[f(\bar{x}_T)] - f(x^*) \;\leq\; \mathbf{E}\left[\sum_{s=1}^{S}\frac{n_s}{T}\left(f(x_s) - f(x^*)\right)\right]$$
$$\leq\; \frac{1}{T}O(GD\sqrt{T}\log(T))$$
$$\leq\; O(GD\log(T)/\sqrt{T})\,.$$

$\square$

### E.3   Proof of Lemma 4.3

*Proof.* Let $S$ be the total number of times that LazySGD invokes the AE procedure. We will first upper bound the expectation of the following sum (weighted regret):

$$\sum_{s=1}^{S} n_s\left(f(x_s) - f(x^*)\right)$$
$$\leq\; \sum_{s=1}^{S} n_s(g_s^{\top}(x_s - x^*) - \frac{H}{2}\|x_s - x^*\|^2)$$
$$\leq\; \underbrace{\sum_{s=1}^{S} n_s(\tilde{g}_s^{\top}(x_s - x^*) - \frac{H}{2}\|x_s - x^*\|^2)}_{(a)} + \underbrace{\sum_{s=1}^{S} n_s(g_s - \tilde{g}_s)^{\top}(x_s - x^*)}_{(b)} \qquad (14)$$

where we have used the $H$-strong-convexity of $f(\cdot)$. The proof goes on by bounding the expectation of terms $(a)$, $(b)$ appearing above.

**Bounding term (a):**   Assume that LazySGD uses the AE procedure with some $\delta > 0$. Since LazySGD is equivalent to SC-AdaNGD$_2$ with $\|g_s\|^2 \leftarrow n_s$ and $g_s \leftarrow n_s g_s$, then a similar analysis to $SC - \text{AdaNGD}_2$ may show that this sum is bounded by $O(\log T)$. For completeness we provide the full analysis here. Consider the update rule of LazySGD: $x_{s+1} = \Pi_{\mathcal{K}}(x_s - \eta_s n_s \tilde{g}_s)$. We can write:

$$\|x_{s+1} - x^*\|^2 \leq \|x_s - x^*\|^2 - 2\eta_s n_s \tilde{g}_s^{\top}(x_t - x^*) + \eta_s^2 n_s^2 \|\tilde{g}_s\|^2$$

Re-arranging the above we get:

$$n_s \tilde{g}_s^{\top}(x_s - x^*) \leq \frac{1}{2\eta_s}(\|x_s - x^*\|^2 - \|x_{s+1} - x^*\|^2) + \frac{\eta_s}{2}n_s^2 \|\tilde{g}_s\|^2$$

Summing over all rounds we conclude that w.p.$\geq 1 - \delta T$:

$$
\begin{aligned}
\textbf{(a)} \;&=\; \sum_{s=1}^{S} n_s \tilde{g}_s^{\top}(x_s - x^*) - n_s \frac{H}{2}\|x_s - x^*\|^2 \\[2mm]
&\leq\; \sum_{s=1}^{S} \frac{\|x_s - x^*\|^2}{2}\left(\frac{1}{\eta_s} - \frac{1}{\eta_{s-1}} - n_s H\right) + \sum_{s=1}^{S} \frac{\eta_s}{2} n_s^2 \|\tilde{g}_s\|^2 \\[2mm]
&\leq\; 0 + 32 m_0^2 \sum_{s=1}^{S} \eta_s n_s \\[2mm]
&\leq\; \frac{32 m_0^2}{H} \sum_{s=1}^{S} \frac{n_s}{\sum_{k=1}^{s} n_s} \\[2mm]
&=\; \frac{32 m_0^2}{H}\Big(1 + \log\big(\sum_{s=1}^{S} n_s\big)\Big) \\[2mm]
&\leq\; \tilde{O}\Big(\frac{G^2 \log T}{H}\log(1/\delta)\Big)\,. \hspace{3cm}(15)
\end{aligned}
$$

here in the first inequality we denote $\eta_0 = \infty$, the second inequality uses $1/\eta_s = H\sum_{i=1}^{s} n_i$, and also $n_s\|\tilde{g}_s\|^2 \leq 64 m_0^2$, which follows by Theorem E.1; the fourth inequality uses Lemma D.2. We also make use of $\sum_{s=1}^{S} n_s = T$.

Since **(a)** is bounded by $2GDT$, then taking $\delta = O(1/T^2)$ ensures that,

$$
\mathbf{E}[\textbf{(a)}] \;\leq\; O(G^2 \log^2(T)/H)\,. \hspace{3cm}(16)
$$

**Bounding term (b):** Similarly the proof of Lemma 4.2 (see Section E.2) we can show that,

$$
\mathbf{E}[\textbf{(b)}] \;=\; 0\,.
$$

Using Jensen's inequality and combining the above with Equations (14) ,(16), establishes the lemma:

$$
\begin{aligned}
\mathbf{E}[f(\bar{x}_T)] - f(x^*) \;&\leq\; \mathbf{E}\Big[\sum_{s=1}^{S} \frac{n_s}{T}\big(f(x_s) - f(x^*)\big)\Big] \\[2mm]
&\leq\; O\left(\frac{G^2 \log^2(T)}{HT}\right)\,.
\end{aligned}
$$

$\square$