[Reviews · NeurIPS 2017]

Reviewer 1



The paper introduces methods for convex optimization which is based on converting adaptive online algorithms into offline methods. Concretely, it presents extensions and generalizaitons of AdaGrad [1] in the case of convex and strongly-convex functions in both offline and stochastic settings. The introduced methods guarantee favorable convergence results in all settings. The paper is well-written and nicely organaized. In particular, the intuitive proof sketches makes it easy to follow/read. The paper provides supportive theoretical analysis and the proofs seem technically sound. The desirable convergence results of the generalizations of AdaGrad (AdaNGD, SC-AdaNGD, LazySGD) look theoretically signficant. TYPOS: LINE 192: well-known

Reviewer 2



This paper considers two striking problems in optimizing empirical loss in learning problems: adaptively choosing step size in offline optimization, and mini-batch size in stochastic optimizations. With a black-box reduction to into AdaGrad algorithm and utilizing gradient normalization and importance weighting in averaging the intermediate solutions, this paper presents a variant of AdaGrad algorithm that attains same convergence rates as GD in convex and smooth optimization without the knowledge of smoothness parameter in setting the learning rate or any linear search (except strongly convex setting, where the algorithm still needs to know the strong convexity parameter H). In stochastic setting, the paper presents a LazySGD algorithm that adaptively chooses the mini-batch size according to the norm of sampled gradients. The paper is reasonably well written, and the structure and language are good. The paper is technically sound and the proofs seem to be correct as far as I checked. The authors also included some preliminary experiments on a toy example to demonstrates some their theoretical insights. Overall, the paper is well written and has some nice contributions.